# Sodium Accumulation in Infected Cells and Ion Transporters Mistargeting in Nodules of *Medicago truncatula*: Two Ugly Items That Hinder Coping with Salt Stress Effects

**DOI:** 10.3390/ijms231810618

**Published:** 2022-09-13

**Authors:** Natalia A. Trifonova, Roman Kamyshinsky, Teodoro Coba de la Peña, Maria I. Koroleva, Olga Kulikova, Victoria Lara-Dampier, Pavel Pashkovskiy, Mikhail Presniakov, José J. Pueyo, M. Mercedes Lucas, Elena E. Fedorova

**Affiliations:** 1Timiryazev Institute of Plant Physiology, Russian Academy of Science, 127276 Moscow, Russia; 2National Research Center Kurchatov Institute, 123098 Moscow, Russia; 3Centro de Estudios Avanzados en Zonas Áridas (CEAZA), La Serena 2940000, Chile; 4Department of Plant Sciences, Wageningen University, 6708 PB Wageningen, The Netherlands; 5Instituto de Ciencias Agrarias (ICA-CSIC), 28006 Madrid, Spain

**Keywords:** symbiosis, ion transport, salt stress, sodium, potassium, NHX Na^+^/K^+^ exchangers, *Medicago truncatula*, root nodule

## Abstract

The maintenance of intracellular nitrogen-fixing bacteria causes changes in proteins’ location and in gene expression that may be detrimental to the host cell fitness. We hypothesized that the nodule’s high vulnerability toward salt stress might be due to alterations in mechanisms involved in the exclusion of Na^+^ from the host cytoplasm. Confocal and electron microscopy immunolocalization analyses of Na^+^/K^+^ exchangers in the root nodule showed the plasma membrane (MtNHX7) and endosome/tonoplast (MtNHX6) signal in non-infected cells; however, in mature infected cells the proteins were depleted from their target membranes and expelled to vacuoles. This mistargeting suggests partial loss of the exchanger’s functionality in these cells. In the mature part of the nodule 7 of the 20 genes encoding ion transporters, channels, and Na^+^/K^+^ exchangers were either not expressed or substantially downregulated. In nodules from plants subjected to salt treatments, low temperature-scanning electron microscopy and X-ray microanalysis revealed the accumulation of 5–6 times more Na^+^ per infected cell versus non-infected one. Hence, the infected cells’ inability to withstand the salt may be the integral result of preexisting defects in the localization of proteins involved in Na^+^ exclusion and the reduced expression of key genes of ion homeostasis, resulting in premature senescence and termination of symbiosis.

## 1. Introduction

Fabacean root nodules are formed as a result of symbiotic relationships with different genera of soil bacteria, generally known as rhizobia. The root nodule contains some special cells (infected cells) serving as a temporary niche for thousands of rhizobia enclosed in a plant-derived membrane, an organelle-like structure, the symbiosome [1,2]. The root nodule becomes a source of nitrogen in the form of ammonia that is produced by differentiated rhizobia (bacteroids). In the process of performing the functions of the host, the plant-infected cell undergoes significant changes, which include the reduction in lifespan, and changes in the expression of different genes, the cytoskeleton pattern, membrane traffic, and distribution of several proteins [3,4]. In connection with these changes, the infected cell may have preexisting features that should be taken into account and require investigation to improve the efficiency of symbiosis and its resistance to harmful environmental conditions.

It is well known that root nodules are more sensitive than the host plant itself to stress caused by factors such as salinity. Salt stress negatively affects nodulation and damages the root nodule structure and ultrastructure, resulting in early symbiosome senescence and a decrease in nitrogen fixation [5,6,7,8]. To improve the nodule tolerance to salt stress, studies involving natural selection and bioengineering of the host plant [7,9,10,11] and the microsymbionts [12,13] have been performed. However, the reasons behind the nodule sensitivity to salt stress are not fully understood.

One of the main effects of salt stress in plants is an induced change in cell homeostasis, caused by Na^+^/K^+^ unbalance. Hence, Na^+^ exclusion constitutes one of the main mechanisms of salt stress responses in plant cells. The efflux of Na^+^ from the plant cell depends on exchangers/antiporters K^+^ (Na^+^)/H^+^ (NHX), which are included in the monovalent cation/proton antiporters (CPA) family [14,15,16,17]. The NHX potassium/sodium exchangers play significant roles in diverse cellular processes—pH regulation, vesicle trafficking and fusion, growth and development, and salt tolerance—and they are considered the main “players” in salt stress response [17,18,19]. According to the extensive studies in *Arabidopsis thaliana*, NHX are capable of transporting either K^+^ or Na^+^ into the vacuole or endosome in exchange for H^+^ efflux to the cytosol [NHX1–6], or facilitating a Na^+^ efflux out of the cell in exchange for a H^+^ influx into the cell (NHX7/SOS1) [17,20,21,22]. The *A. thaliana* genome contains eight *NHX* genes. *AtNHX1*, *AtNHX2*, *AtNHX3*, *AtNHX4*, and *AtNHX6* are expressed in roots, shoots, and seedlings, and *AtNHX7* (*SOS1*) is mainly expressed in roots and induced by salt stress [23,24,25].

The maintenance of cellular ion homeostasis in plants also involves K^+^ transporters of the group of KT-HAK-KUP, which include the HAK (high-affinity K^+^) and the KUP/KT (K^+^ uptake permease/K^+^ transporter) families that contribute to K^+^ homeostasis in low and high external concentrations, respectively [26,27]. Plants also have a large family of predicted cation/H^+^ exchangers (CHX) with functions not yet fully defined, which may also be involved in pH and cation homeostasis [15,16].

Less is known about the gene expression, location, and functions of the exchangers/antiporters in *Medicago truncatul**a* and the information concerning their functionality in root nodules is scarce. The *M. truncatula* genome contains six NHX genes; *MtNHX1*, *MtNHX6*, and *MtNHX7* (Medtr2g038400, *Chr2*) that display high expression in roots and leaves; *MtNHX2* and *MtNHX3* with the highest expression in flowers and relatively low expression levels in other tissues, and *MtNHX4* with the highest expression in pods [28]. Upregulation of *NHX7/SOS1* has been associated with salt tolerance in *M. truncatula* and *M. falcata* and other Fabaceae [28,29,30]. A decrease in K^+^ levels in symbiosomes and vacuoles during the infected cell lifespan, and the mistargeting and partial depletion from the plasma membrane of mature infected cells of two plant Shaker K^+^ channels—inward-rectifier K(^+^) channel AKT1 and potassium outward channel SKOR/GORK in mature infected cells—might also compromise the K^+^ availability in both the host cell and the symbiosome [31].

With the aim to clarify the mechanisms involved in Na^+^ exclusion and K^+^ transport in root nodule cells of *M. truncatula* subjected to salt stress, we performed: (1) in silico expression analysis of selected ion transporters/channels/exchangers genes of both the host plant and the microsymbiont in nodules; (2) the immunolocalization of Na^+^/K^+^ exchangers MtNHX6 and MtNHX7 in root nodules; (3) qPCR analysis of selected genes involved in ion translocation; and, (4) low temperature-scanning electron microscopy (LTSEM) and X-ray microanalysis of *M. truncatula* nodules from plants exposed to salt stress and control plants. Our results suggest that the proteins putatively involved in the exclusion of Na^+^ from the cell cytoplasm in mature infected cells are being lost from their target membranes and expelled to young vacuoles. A downregulation (and even suppression of transcription) of genes related to Na^+^ efflux and K^+^ inward transport occurs in the internal region of the root nodule, making the mature infected cells less protected against salt stress. The combination of these events results in excess accumulation of Na^+^ in infected cells and symbiosomes, thus hampering the ability of infected root nodule cells and the root nodule itself to withstand salt stress.

## 2. Results

### 2.1. In Silico Gene Expression Analysis of Ion Transporters, Channels, and Exchangers Genes in Root Nodule Developmental Zones

The expression of genes encoding ion transporters, channels, and exchangers of sodium and potassium of the host plant and intracellular rhizobia putatively involved in salt stress defense mechanisms in the nodule was estimated by in silico gene expression analysis in the nodule developmental zones using the Symbimics portal [32,33]. This portal contains gene expression data obtained by laser dissection of the cells of the defined developmental zones of the wild-type mature *M. truncatula* nodule. Developmental zones of root nodules: meristem, infection zone (ZII), where rhizobia are released into the plant cell and start their differentiation process, the interzone II/III (IZ), and the nitrogen fixation zone (ZIII), which is the mature part of the nodule consisting of small non-infected cells and large infected cells that contain the differentiated nitrogen-fixing bacteroids. The basal part of the nodule contains the zone of senescence where the symbiosis is terminated (ZIV).

The expression of *M. truncatula* and *S. meliloti* genes in the nodule developmental zones are shown in Table 1 and Table 2, respectively, accession numbers listed in Appendix A. A substantial number of the analyzed genes showed changes in their expression level in the developmental zones of the nodule, suggesting important adaptations/alterations in ions transport in both symbionts during the development of symbiosis. The spatial expression analysis of the plant genes showed a tendency to downregulation rather than upregulation of the host plant genes in the nitrogen-fixing part of the nodule, as compared with the meristem. All selected genes were expressed in the nodule meristem, except for *MtNHX2* and *MtNHX3*, which were not expressed in either roots or nodules (Table 1). Conversely, in the mature part of the nodule (ZIII), where bacteroids fix nitrogen, 7 of the 20 analyzed genes were either not expressed or substantially downregulated. These genes included the homologs of *A. thaliana* genes *HAK5*, *KAT1*, and *NHX7*, which are important in salt stress response. In ZIII, six genes were upregulated (*MtHAK6*, *MtHAK7*, *MtSKOR/GORK*, *MtCHX18*, *MtNHX1*, *MtNHX6*). The upregulation of *MtNHX1* and *MtHAK6* was also reported by Limpens et al. [34] Appendix A).The expression of *MtKT3* and *MtNHX8* was stable in all nodule zones, and the rest of the genes showed a downregulation with respect to the meristem (Table 1).

These results indicate that the K^+^/Na^+^ transport systems involved in Na^+^ sequestration in the mature part of the nodule might be compromised. The expression of rhizobial genes involved in the inward transport of K^+^ and the sodium/proton antiporter Na^+^/H^+^
*NhaA* showed upregulation in ZIII as compared with the gene expression levels in the infection zone (FIId, FIIp) (Table 2). However, such an increase probably reflects the drastic increase in the rhizobia number per host cell in mature infected cells that were up to 25 times higher in comparison with the host cell at the start of infection [35].

### 2.2. Localization of MtNHX7 and MtNHX6 in Root Nodule Cells by Confocal and Electron Microscopy

Two *M. truncatula NHX* exchangers, MtNHX7 and MtNHX6, were used for immunolocalization analysis. *MtNHX7* is a homolog of the *AtNHX7 (SOS1)* gene. AtSOS1 is located on the plasma membrane and involved in Na^+^ exclusion from the plant cell via the plasma membrane to the apoplast. The upregulation of *NHX7/SOS1* has been associated with salt tolerance in *M. truncatula* and *M. falcata* [28,29], and other fabacean species [30]. The exchanger MtNHX6 is a homolog of AtNHX6 [20]. AtNHX6, a Golgi/endosome/PVC resident, participates in K^+^ or Na^+^ sequestration into endosomal lumen [17,25]. This endosomal exchanger has been selected for the immunolocalization study due to the known changes in the symbiosome membrane identity, and the appearance of some PVC endosome markers on the membrane during the symbiosis development [35]. Wild-type and transgenic nodules carrying the *ProMtNHX7:MtMTNHX7:GFP* construct were used for the localization study of MtNHX6 and MtNHX7, using anti-MtNHX6 and anti-GFP antibodies, respectively. A secondary Alexa 488-conjugated antibody was used in both cases.

The MtNHX7 protein signal was present in the plasma membrane (PM) of cortex cells, vascular bundles, meristematic cells, and young nodule cells in the infection zone (Figure 1A–D). In mature infected cells, the pattern of the protein localization was altered, and the signal was lost from the PM and was detected in the cytoplasm, forming a dot-like pattern that reflects a process of internalization of the protein from the PM (Figure 1C).

The localization analysis of MtNHX6 in nodules showed the presence of the protein in the cytoplasm (Figure 2A), especially strong on the tonoplast membranes (Figure 2B), reflecting the fusion of endosomes with the tonoplast membrane in the meristem and in newly infected cells (Figure 2B). At high magnification, the immunosignal was discernible in small bodies with the size of endosomes/vesicular bodies budding from the trans-Golgi network (50–100 nm) (Figure 2C). At the tissue level, the signal was observed in meristematic cells, distal and proximal infection zones, and in vascular bundles (Figure 2D,E). In the mature infected cells, however, the pattern of protein distribution was changed, the signal was observed over big dots (500–1000 nm), and young lytical vacuoles, which may suggest the process of mistargeting and the depletion of the protein from its target membranes. The presence of the immunosignal, labeled by gold particles, in vacuole-like bodies (500–1000 nm) was confirmed by electron microscopy, the signal was present in the lumen of young vacuoles (Figure 3A,B).

### 2.3. Ion Compartmentalization Analysis

Ion compartmentalization analysis was performed in the mature nitrogen-fixing part of root nodules. High magnification was used to distinguish individual symbiosomes and vacuoles in nodules from control plants and from plants subjected to salt stress (Figure 4A). The distribution of sodium (Na), potassium (K), magnesium (Mg), phosphorus (P), sulfur (S), chlorine (Cl), calcium (Ca), iron (Fe), copper (Cu), and molybdenum (Mo) was analyzed. The content and distribution of sodium (Na) and potassium (K) are presented in Figure 4B,C; the data of other elements are presented in Appendix A. In nodules subjected to salt treatment, an increase in Na^+^ content ‘on spot’ was statistically significant in the cytoplasm of both infected and non-infected cells, and in bacteroids (Figure 4B) as compared with nodules of control plants. K^+^ content diminished 2–3 times in the vacuoles of non-infected cells (Figure 4C).

Concerning other ions, salt stress significantly decreased the contents of Ca^2+^ in the vacuoles of non-infected cells, and of S in the vacuoles of infected cells. The chlorine content increased in the vacuoles and in the cytoplasm of non-infected and infected cells, and the content of Fe increased in the cytoplasm of infected cells (Appendix A). According to the results, in the mature part of the nodules of salt-treated plants, Na^+^ accumulated in the cytoplasm and in the bacteroids, while there was a decrease of K^+^ content in the vacuoles. These results suggest that in salt-stressed nodules, infected and non-infected cells presented an insufficient ability to expel Na^+^ from the cytoplasm and to prevent Na^+^ accumulation in the symbiosomes.

### 2.4. qPCR Expression Analysis of Nodules under Salt Stress

With the aim to detect the effect of salt stress on the expression of plant genes involved in the exclusion of Na^+^ and translocation of K^+^, we analyzed nodules from salt-treated and control plants. The effect of the salt treatment on the expression of genes related to the exclusion of Na^+^ was analyzed: *MtNHX1* and *MtNHX6*, involved in sequestration of Na^+^ into vacuoles and endosomes, *MtNHX7*, the homolog of *SOS1* of *A. thaliana* involved in Na^+^ efflux out of the cell via the plasma membrane to the apoplast, and recently annotated *MtNHX7-like* and *MtNHX8* genes, as putative homologs were evaluated (Figure 5A). Salt stress led to significant upregulation of *MtNHX1*, *MtNHX7*, and *MtNHX7*-like genes. Genes involved in the translocation of K^+^ were also investigated: The expression of the high-affinity potassium transporter *MtHAK7,* the H^+^/K^+^ exchanger *MtCHX18*, the potassium inward channel *MtAKT1*, and the potassium outward channel *SKOR/GORK* were not affected by salt; however, the expression of *MtHAK6* was significantly downregulated (Figure 5B). The expression analysis of the *nifH* gene encoding one of the subunits of nitrogenase [6] showed a tendency to downregulation that can be expected as a well-known effect of salt treatment on nitrogen-fixing activity (Figure 5C). In summary, salt stress induced certain genes from the NHX clade, involved in Na^+^ sequestration, as could be expected, but had no effect on the expression of K^+^ translocators genes, other than the downregulation of *MtHAK6*.

## 3. Discussion

Root nodule infected cells, which are housing thousands of symbiosomes, are physiologically different from non-infected cells. They have a short lifespan not exceeding four weeks and other special biological features, such as the modification of the endomembrane system, the transformation of the cytoskeleton pattern, and changes of vectors and destination membranes in membrane trafficking [2,4,35,36,37,38]. One of the most important features of infected cells is the change in vesicular traffic from the plasma membrane, endosomes, and tonoplast toward the symbiosome membrane, the interface surrounding the bacteria in the host cytoplasm [2,3,4]. Such cell-specific modification of some proteins targeting infected cells favors the propagation and maintenance of the intracellular microsymbiont colony [2,3,35,36,39]. However, the presence of a huge bacterial colony inside the host cells, as it seems, has some negative effects that now start to be revealed. Among the examples of these negative effects, we can indicate the partial loss of functionality of the infected cell vacuoles, manifested by the loss of acidic pH, and a great part of the volume [35] and the drastic changes in the distribution of channel proteins involved in K^+^ transport (AKT1, SKOR/GORK) [31] in mature infected cells. The specific environment of the infected cell, caused by the presence of an intracellular bacterial colony, may have other consequences, making the root nodule more vulnerable to salt stress than other plant organs. Here, we report the location of the MtNHX6 and MtNHX7, which supports the above hypothesis.

In plants, salinity causes osmotic stress at the tissue and the cellular level and decreases turgor pressure, which results in a competition between Na^+^ and K^+^ for the major binding sites in key metabolic processes, affecting enzymatic reactions, protein synthesis, ribosome functions, and cell metabolism [40,41]. Salinity has also a negative effect on photosynthesis [42] which affects root nodules, they consume more than 25% of the total photosynthates [43].

The NHX group exchangers, which are involved in the efflux of Na^+^, are important regulators of endomembrane K^+^/Na^+^ homeostasis due to the process of Na^+^ efflux in response to salt stress [29,44,45,46,47]. Apoplastic Na^+^ depolarizes the plasma membrane, making K^+^ uptake by potassium channel *AKT1* thermodynamically impossible, and activating the gated outward rectifying *K^+^ SKOR*/*GORK* efflux channels [46] which leads to a diminishing of K^+^ content in the cells. The loss of K^+^ is also one of the symptoms of stress caused by pathogens elicitors, reactive oxygen species (ROS), salinity, and drought [48,49].

Similar to other plant tissues, the root nodule responds to the changing environment by adapting to the dynamic endomembrane system and the transport of necessary substances, including ions. The root nodule tissue similar to other plant cells is equipped with the mechanisms to adapt or diminish the negative effects of salt stress. However, the high sensitivity of the root nodule to salinity is an undeniable fact manifested as rapid senescence of the symbiosomes, drop in nitrogen fixation, and termination of symbiosis [5]. In the present paper, we describe a new aspect of symbiosis development: the changes in the localization of NHX exchanger proteins in the root nodule and the response of nodule cells to the salt treatment in terms of ion distribution and gene expression of ion exchangers, channels, and transporters.

Some of the intrinsic features of the root nodule, according to in silico gene expression analysis (Table 1), is the downregulation or termination of expression of several genes involved in the maintenance of optimal cytosolic K^+^/Na^+^ ratio in the mature part of the nodule in comparison with the apical part of the nodule, including the meristem and the infection zone (Table 1). The differential expression of cation transporters in infected and non-infected cells of root nodules was reported before [34] (Appendix A). Genes encoding some metal ion transporters (Cu, Zn), show high expression in non-infected cells, but Mn^2+^ and Fe^2+^ transporters of the NRAMP family and Ca^2+/^H^+^ antiporter VCX1 were highly expressed in infected cells of the zone of nitrogen fixation (34, Appendix A). According to gene expression, data obtained by laser-dissection analysis of infected and non-infected cells in nodule developmental zones, *MtNHX1* and *MtHAK6*, were upregulated in infected cells in the zone of nitrogen fixation. The expression data from the Gene Atlas portal for MtNHX6 and MtNHX7-like genes (Mtr.13430.1.S1_at [log2_rma]) show high expression in young nodule cells of zone II, but the moderate expression in non-infected cells and non-infected cells of nitrogen fixation zone [50].

We speculate that the hypoxic conditions in the mature part of the nodule might be a cause for the spatial differences in gene expression. In root nodules, the central part and especially infected cells have a low O_2_ pressure that facilitates nitrogenase activity [36], but in the root nodule meristem and the cortex, the level of oxygen is not so limited. As it was shown for nodules of alfalfa and sweet clover, both higher O_2_ permeability and higher respiratory capacity are associated with the nodule tip, which contains meristem and zone of initial infection [51]. It has been reported that the expression of some ion channels is downregulated in hypoxic conditions, for example, ion channel GORK in *A. thaliana* roots was downregulated up to 3 times within 1 h of hypoxic treatment [52]. The negative effects of hypoxia on the expression of ion channels are being also intensively studied in animal cells [53].

In *A. thaliana*, Na^+^ influx into the cell is counterbalanced by the active removal of this ion by the mechanisms of Na^+^ extrusion via plasma membrane by AtNHX7/SOS1 [15], or the sequestration of Na^+^ in the vacuole by AtNHX1 and AtNHX2 and in endosomes by AtNHX5 and AtNHX6 [15,20,22,23]. In root nodules, the functional status of these genes may be impaired. According to our data, the spatial pattern and the correct location of exchangers MtNHX6 and MtNHX7 were maintained in nodule meristem, in very young, infected cells, and in most of the non-infected cells. However, in mature nitrogen-fixing cells, exchanger proteins were mistargeted, removed from their destination membranes, and expelled to young vacuolar compartments. The immunolocalization analysis of MtNHX7 and MtNHX6 proteins evidenced comparable dynamics in protein relocation in mature infected cells despite the differences in gene expression in the nodule developmental zones, and their diverse destination membranes: plasma membrane for MtNHX7 and endosome/tonoplast for MtNHX6. Similar changes in localization patterns have also been reported for the MtAKT1 protein involved in K^+^ transport [31] in both mature and senescent infected nodule cells. The reasons for the changes that we observed in the pattern of NHX proteins localization cannot be unequivocally explained. The mistargeting of the proteins might indicate initial signs of degradation of the infected cells or it may be associated with hypoxia in these cells. However, the change in the pattern of ion exchanger localization may be one of the reasons for the low tolerance of infected cells toward the accumulation of Na^+^. It could be expected that the loss of the proteins from their target membranes may be detrimental to the functional activity of MtNHX7 and MtNHX6, specifically in infected cells, which may result in diminished ability to expel or sequester Na^+^ ions and likely affect other combined ion transport pathways. The efficiently functioning MtNHX7 and MtNHX6 proteins seem to be spatially restricted to non-infected cells of the vascular bundles, meristem, and the external cortex where the proteins display a proper localization pattern (Figure 1). The expression of genes found to be upregulated according to qPCR analyses after NaCl application is probably also spatially restricted to these tissues. On the contrary, the hypoxic central nodule tissue, especially infected cells, seems to be less protected against Na^+^ accumulation due to the loss of proteins from the target membranes.

According to the data presented in this paper, mature infected cells accumulate more Na^+^ than non-infected interstitial cells, as confirmed by the ion distribution analysis. The salt treatment led to an increase in Na^+^ content ‘on the spot’ approximately four times in the cytoplasm of infected and non-infected cells, and three times in bacteroids (Figure 3) as compared to the cells of control nodules. We should keep in mind that mature infected host cells, due to the drastic increase in the number of symbiosomes, have a volume that is more than five times that of non-infected cells [35]. That means that, according to IDA analysis, an infected cell accumulates at least 5–6 times more Na^+^ per cell than a neighboring non-infected one, which supports the assumption of the insufficient ability of mature infected cells to remove excess Na^+^ and prevent its accumulation in the bacteroids.

In general, the effect of salinity stress in the central part of the root nodule would be similar to combined waterlogging and salinity stress modeled in *A. thaliana* roots [54,55]. Combined hypoxic conditions and salinity in *A. thaliana* usually lead to more damage to plant growth than salinity stress alone [54]. According to Wang et al. [54], the response to combined hypoxia and salt stress in *A. thaliana* is regulated by RBOHD (respiratory burst oxidase homolog protein D). In salt-treated plants, the onset of hypoxia leads to increased uptake of Na^+^ and Cl^−^ in RBOHD mutant plants, linking the regulation of salt stress response to reactive oxygen species (ROS), produced in the hypoxic environment, which causes the disturbances in K^+^ homeostasis [54,55,56]. To confirm the model in salt-stressed nodules, further analyses of RBOHD are needed.

Although our study was focused on a fraction of the ion exchangers involved in Na^+^ and K^+^ transport in nodules our results reflect the changes that may be attributable to other proteins in the cells housing bacteria. The change in protein location in infected cells may possibly also affect the functionality of other proteins involved in the transport of necessary compounds belonging to other pathways. Such a situation should be considered a preexisting norm for these cells, taken under consideration in bioengineering works.

In conclusion, in *M. truncatula* nodules, the cells of the mature infected zone, containing the nitrogen-fixing bacteria, have a limited capability to withstand salt stress. The reasons for this are the inefficient mechanisms of Na^+^ exclusion as a consequence of the mistargeting of the proteins involved in Na^+^ exclusion in mature nitrogen-fixing cells and the partial downregulation of relevant genes in the mature part of the nodule, leading to the accumulation of Na^+^ especially in the infected cell and in symbiosomes and to the premature termination of symbiosis.

## 4. Materials and Methods

### 4.1. Plant Material, Growth Conditions, and Treatments

*Medicago truncatula* Gaertn. cv. Jemalong A17 plants were grown and transformed by *Agrobacterium rhizogenes* strain MSU440 according to Limpens et al. [57]. *Sinorhizobium* (*Ensifer*) *meliloti* strain Sm2011 or *S. meliloti* Sm2011-mRFP expressing the red fluorescent protein were used for nodulation [58].

*M. truncatula* seeds were sterilized and germinated. Seedlings were sown in vermiculite supplemented with Fahraeus medium and inoculated with *S. meliloti* 2011 3 d after sowing. Plants were grown under long-day conditions (16 h:8 h, day:night) at 20 °C.

Eight-week-old plants were subjected to salt stress by watering with a nutrient solution containing 100 mM NaCl for two weeks. For the analysis, we used nodules from the second generation, which was elicited on lateral roots around 4 weeks later than the first generation that is formed on the main root. These nodules did not display a visible senescent zone.

### 4.2. In Silico Gene Expression Analysis of S. meliloti and M. truncatula Ion Transporters, Channels, and Exchangers Putatively Involved in Salt Stress Response

The expression level of putative *M*. *truncatula* and *S. meliloti* exchangers and K^+^ transporters and channels was estimated in the different nodule developmental zones using the Symbimics portal database (https://iant.toulouse.inra.fr/symbimics/, (accessed on 2 September 2022)) [33,34] obtained by laser-capture microdissection of the root nodule zones and infected and non-infected cells.

*Arabidopsis thaliana* genes were retrieved from the available genomic and cDNA sequence databases (https://www.uniprot.org, (accessed on 2 September 2022)) and used as query sequences. Selected protein sequences were used for the search of *M. truncatula* homologs in public bioinformatic resources (https://phytozome.jgi.doe.gov/pz/portal (accessed on 2 September 2022), https://www.ncbi.nlm.nih.gov/ (accessed on 2 September 2022)). Accession numbers or gene identifiers used in this study are given in Appendix A. Recently, two new exchanger homolog genes found in the *M. truncatula* Annotation Release 102 (MtrunA17r5.0, February 2021) were referred to in the NCBI database, that is, “NHX7-like” and “NHX8”. In this study, we name them *MtNHX7*-like and *MtNHX8*, respectively, and we maintain *MtNHX7* for the gene reported earlier [28]. To avoid misunderstanding, we follow the established nomenclature proposed earlier [28], but their equivalences in several databases are shown in Appendix A for consultation.

### 4.3. RNA Extraction and qPCR Gene Expression Analysis

Total RNA was isolated from control and treated nodules using the RNeasy PlantMinikit (Qiagen, Hilden, Germany). The reverse transcription reaction was performed using the High-Capacity cDNA Reverse-Transcription Kit (Applied Biosystems, Foster City, CA, USA). Gene-specific primers were designed using the PRIMER 3-PLUS software [59] and the Primer Express v3.0 software (Applied Biosystems, Foster City, CA, USA). The primers used are listed in Appendix A. The gene identifiers or locus names of the studied genes are shown in Appendix A. Gene annotation is according to the NCBI *Medicago truncatula* Annotation Release 102 (February 2021).

The expression analysis of NHX exchangers *MtNHX1*, *MtNHX6*, *MtNHX7*, *MtNHX7*-like, *NHX8*, *CHX* exchanger *MtCHX18*, potassium channels and transporters *MtAKT1*, *MtSKOR/GORK*, *MtHAK6*, *MtHAK7*, and nitrogenase subunit *NifH* was performed by real-time qPCR using a 7300 Real-Time PCR System (PE Applied Biosystem, Foster City, CA, USA) with SYBR Green Supermix (Bio-Rad, Hercules, CA, USA), primers are listed in the Appendix A). The constitutively expressed *Mtc27*, *MtGAPDH,* and *SMc00128* genes were used as endogenous controls [6,60]. The comparative C_T_ method [61] or the standard curve method [62] were applied for relative quantification. Primers are listed in the Appendix A.

### 4.4. Cloning

The *MtNHX7* open reading frame and its 2.5-kb regulatory sequence were amplified via PCR from nodule cDNA and genomic DNA, respectively, using Phusion high-fidelity polymerase (Finnzymes). The coding sequence of *MtNHX7* was directionally cloned into a modified pENTR vector (pENTR2) containing a multiple cloning site. Entry clones for *MtNHX7* promoters were generated by TOPO cloning (Invitrogen, Waltham, MA, USA). The Gateway cloning system (Invitrogen) was used to create a genetic construct with the GFP fusion [59]. The pENTR clone of *MtNHX7* was recombined into the pKGW-GGRR destination vector using LR Clonase (Invitrogen): The *ProMtNHX7:MtNHX7:GFP* translational fusion driven by the 2.5 kb native 5′ regulatory sequences with its own promoter was generated by multiple cloning of gene and promoter sequences into the pKGW-MGW vector. Primers are listed in Appendix A. Gene identifiers of *MtNHX6* (*Medtr2g028230*) and *MtNHX7* (*Medtr2g038400*), respectively. Primers are listed in the Appendix A.

### 4.5. Confocal Microscopy

A Leica MZFLIII fluorescence microscope with filter cubes for GFP (excitation, 470/40; Dichroic: 495; emission, 525/50) and DsRed (excitation, 545/30; emission, 620/60) was used for the selection of transgenic tissue. Minimal 10 nodules per trial were fixed in 1% formaldehyde, cut along the longitudinal axis, and blocked with 2% BSA. The GFP signal in the sections carrying the *ProMtNHX7:MtNHX7:GFP* construct was enhanced by using an antibody (Ab) against GFP developed in rabbit (Molecular Probes) at 1:50 dilution. For immunolocalization of MtNHX6, a custom-made antibody was developed in rabbit by GenScript (Piscataway, NJ, USA) from the peptide sequence SENEISPADVHKAP (peptide-KLH conjugate) and used in a 1:100 dilution. The antibody recognized a protein of 58 kDa (Appendix A). An anti-rabbit Alexa 488 (excitation max. 490, emission max. 525 nm; Molecular probes), diluted 1:200, was used as a secondary antibody. A mix of 0.5% skim-milk powder with 2% BSA was used as a blocking solution. Sections were counterstained with 0.01% propidium iodide (PI). Confocal imaging was done on hand-sectioned nodules with a Meta LSM 510 microscope (Leica) in a single image mode.

### 4.6. Electron Microscopy

Transgenic nodules carrying the *ProNHX7-MtNHX7:GFP* construct were fixed and embedded in Lowicryl K4M [3,35]. Ultrathin sections were deposited on nickel grids, blocked with 2% BSA, and left overnight at 4 °C in anti-GFP primary antibody developed in rabbit (Molecular Probes) at 1:50 dilution, then washed 3 times with PBS, 30 min each change, followed by Colloidal 10 nm gold-conjugated Goat anti-Rabbit Ab (British BioCell International (BBI) (Ted Pella Inc., Redding, CA, USA) 1:100 dilution, 2 h, 4 °C. For the localization of MtNHX6 a custom-made Ab developed in rabbit (GenScript, Piscataway, NJ, USA) was used at dilution of 1:50, followed by Colloidal 10 nm gold-conjugated Goat anti-Rabbit Ab (British BioCell International (BBI)(Ted Pella Inc.,Redding, CA, USA). Ultrathin sections were counterstained by 2% uranyl acetate and Reinolds lead citrate [3,35], and analyzed with a LIBRA120 (Carl Zeiss Microscopy, Jena, Germany) electron microscope.

### 4.7. Ion Distribution Analysis (IDA)

IDA was performed by low-temperature scanning electron microscopy (LTSEM) and energy-dispersive X-ray microanalysis (EDXRMA) as described before [31]. Scanning electron cryomicroscopy (cryo-SEM) was performed with a Versa 3D DualBeam (FEI, US), equipped with an energy dispersive X-ray (EDX) spectroscopy system (EDAX, Mahwah, NJ, USA) and a Quorum Tech PP3010T cryogenic system (Quorum Tech, Laughton, East Sussex, UK). Prior to the study, the nodules were glued to the metal holders, then specimens were frozen in supercooled liquid nitrogen and transferred into the Quorum Tech PP3010T preparation chamber under vacuum, where nodules were fractured. Afterward, the specimens were transferred to the cryo-stage cooled by a continuous nitrogen flow at −140 °C. IDA analysis was performed in a spot acquisition mode using an accelerating voltage of 20 kV, 4 nA current, and a resolution of 133 eV. Quantitative element analysis was obtained using standard ZAF (atomic number, absorption, and fluorescence) correction procedures with TEAM 4.4 (EDAX, Mahwah, NJ, USA) and with Link Isis, version 3.2 (Oxford, UK) For each analysis, 10 nodules were used, 10–15 cells were analyzed per nodule, and 7–12 symbiosomes and vacuoles were examined.

### 4.8. Western Blot Analysis

Proteins were extracted from root tips in 25 mM Tris-HCl buffer containing 1 mM EDTA, 1 mM DTT, and protease inhibitors cocktail (Roche, Basel, Switzerland). Sixty μg of protein were loaded per well. Proteins were separated by 12% sodium dodecyl sulfate-polyacrylamide gel electrophoresis and were blotted onto a nitrocellulose membrane (Bio-Rad). The membrane was incubated in 3% BSA, as a blocking agent, followed by a primary anti-NHX6–specific antibody, 1:50 dilution, followed by the secondary antibody, an anti-rabbit immunoglobulin G peroxidase produced in goat (Sigma, Saint Louis, MI, USA), 1:5000 dilution. The immunosignal was visualized by incubation with the Immuno-Star Western chemiluminescent kit (Bio Rad, Hercules, CA, USA). The antibody recognized a protein of 58 kDa (Appendix A). Blots were photographed with a Molecular Imager Chemi Doc XRS+, using Image Lab software in Chemi mode. The pre-stained protein ladder was photographed in normal light.

### 4.9. Phylogenetic Analysis

Nucleotide sequences of *A. thaliana* AtNHX1-8 homologs in *M. truncatula* and other legume and non-legume species were retrieved from several genome sequence databases and plant genomic resources: National Centre for Biotechnology Information (https://www.ncbi.nlm.nih.gov/ (accessed on 2 September 2022)), Phytozome (https://phytozome.jgi.doe.gov/pz/portal.html (accessed on 2 September 2022)), and the Legume Information System (https://legumeinfo.org/ (accessed on 2 September 2022)). *A. thaliana* AtNHX1-8 sequences were used as queries.

Phylogenetic analysis was inferred using the neighbor-joining method [63,64]. The evolutionary distances were obtained by the maximum composite likelihood method [65]. Node robustness was assessed by the bootstrap method (*n* = 1000 pseudo-replicates). Evolutionary analyses were conducted in MEGA X [61]. Phylogenetic Analysis of *M. truncatula* Homologs of the NHX Group.

Due to the important role of NHX exchangers in plant ion homeostasis, the various NHX genes recently annotated in the *M. truncatula* genome, the high expression of certain members of this group in the mature part of the nodule, and the upregulation of certain NHX genes found in salt-stressed nodules, we performed a phylogenetic analysis of these genes (especially focused on NHX6-7-8 genes) to establish homologies with *A. thaliana* AtNHXs and other functionally characterized homologs from other plant species (*Oryza sativa*, *Phoenyx dactylifera*, *Triticum aestivum*, *Beta vulgaris*, *Populus trichocarpa*). Moreover, phylogenetic relationships with NHX6-7-8 homologous genes from other legume species were also characterized.

The phylogenetic analysis showed that NHX homologs clustered into three mean clades (Figure 6). Nodes in most clades displayed strong support, with bootstrap values of 72 or higher. The first main clade (where only *A. thaliana* and *M. truncatula* genes were included) contains NHX1-4 genes from *A. thaliana* and their corresponding homologous from *M. truncatula*. MtNHX1 and MtNHX2 genes were closely associated with AtNHX1 and AtNHX2, all of them included in the same clade. MtNHX3 and MtNHX4 are also closely associated with AtNHX3 and AtNHX4. This clade topology agrees with that obtained by [28].

In the second main clade, MtNHX6 clustered together with previously functionally characterized homologs from the included species *A. thaliana* (AtNHX5 and AtNHX6) [20,28], *Phoenix dactylifera* [66], *Oryza sativa* [67], *Beta vulgaris* [68], and *Populus trichocarpa* [69], suggesting similar functions for MtNHX6 as endosome-associated antiporters involved in ion homeostasis in vacuoles. MtNHX6 and all other legume genes clustered together in a clade, according to a classical legume phylogeny (Figure 6). It is interesting to note that *M. truncatula* and the other Hologalegina legumes have one NHX6-like homolog as is also the case for Genistoids (*Lupinus angustifolius*). However, Mimosoid and Millettioid legumes have at least two NHX6 homologs. The cultivated peanut, the Dalbergioid, *Arachis hypogaea*, displays two homologs, probably due to its origin from the hybridization of *A. duranensis* and *A. ipaensis*, both of them containing only one NHX6 homolog (not shown). The third main clade included NHX7 and NHX8 homologs of *M. truncatula*, *A. thaliana*, and other legume and non-legume species (Figure 6). We included the MtNHX7-like and MtNHX8 new genes reported in the NCBI Medicago truncatula Annotation Release 102 (Assembly Mtru-nA17r5.0-ANR). Both genes were closely associated with MtNHX7, AtNHX7, and AtNHX8 homologs. These genes clustered together with functionally characterized homologs from the included species *Oryza sativa* [70], *Triticum aestivum* [71], *Beta vulgaris* [68], *Populus trichocarpa* [69], and the legume *Glycine max* [72], suggesting the involvement of MtNHX7 in salt tolerance. *M. truncatula* genes and all other legume genes clustered together in a clade, according to a classical legume phylogeny. In this case, Hologalegina species display one, two, or three (for *M. truncatula*) homologs. The Genistoid *L. angustifolius* has two homologs. The Mimosoid Prosopis alba has four homologs, and most Millettioids display one homolog (except *Cajanus cajan*, containing two homologs) (Figure 6).

### 4.10. Statistical Methods 

IDA analysis was performed twice in two independent experiments. Ten nodules from different plants per experiment were used. Minimal ten infected and ten non-infected cells per nodule were analyzed. Gene expression analysis was performed twice in two independent experiments. The Student’s *t*-test was used to determine the significance of the differences between the means of data sets.

## Figures and Tables

**Figure 1 ijms-23-10618-f001:**
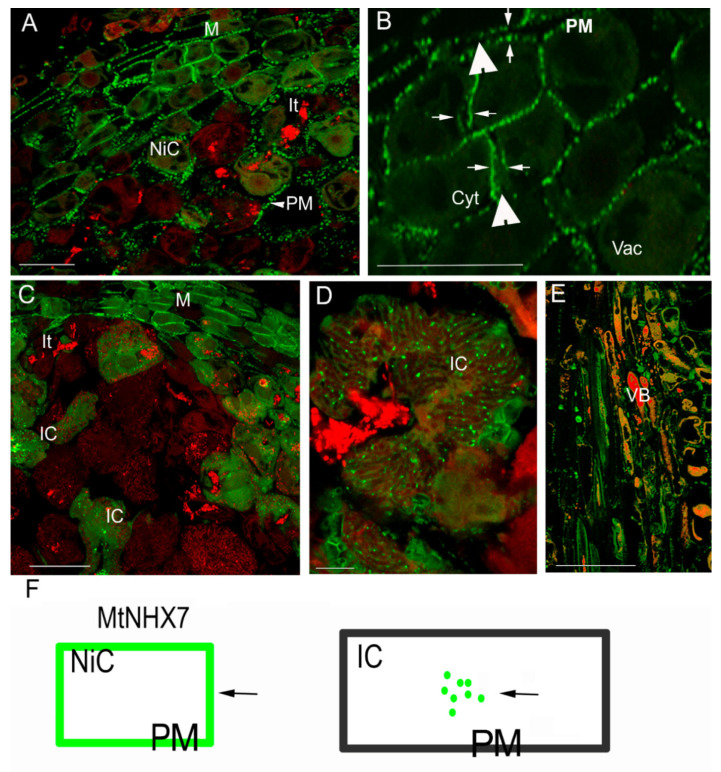
Confocal microscopy immunolocalization of the MtNHX7 protein in the transgenic nodules carrying the *ProMtNHX7:MtMTNHX7:GFP* construct (**A**–**E**), and the scheme of protein distribution in infected and non-infected cells as per immunosignal (in green color) to explain the localization pattern and its changes (**F**). Color code: red fluorescence—propidium iodide counterstained bacteria and nuclei, green fluorescence—protein labeling. MtNHX7 distribution was visualized by GFP signal in the sections carrying the *ProMtNHX7:MtNHX7:GFP* construct. Green fluorescence in the nodule meristem and distal infection zone (close to the meristem) was detected in the plasma membrane (arrowhead). (**B**) High magnification of fraction of (**A**). Note the immunosignal is distributed in double rows decorating the plasma membranes of neighboring cells (arrows), separated by the cell wall (arrowhead). (**C**) Distal infection zone and fully infected cells situated below the infection zone, MtNHX7 is detected in the plasma membrane in meristematic cells and in cells containing freshly released bacteria. (**D**) Infected cells with MtNHX7 signal present in a dot-like pattern in the cytoplasm. (**E**) Localization of MtNHX7 in the vascular bundles; note the strong membrane signal. (**F**) Scheme of protein distribution in infected and non-infected cells. The arrow is pointing to the green immunosignal. Green rectangular box is representing the plasma membrane (PM) of non-infected cell, that retained the immunosignal, black rectangle is representing the PM of infected cell, with no green signal present, the green labelled protein is located in the cytoplasm in do-like pattern. IC: infected cell, NiC: non-infected cell, M: meristem, It: infection thread, Vac: vacuole, VB: vascular bundle, T: tonoplast. Arrowhead: signal over the plasma membrane (PM). Bars: (**A**,**B**,**E**): 50 μm; (**C**): 25 μm; (**D**): 10 μm.

**Figure 2 ijms-23-10618-f002:**
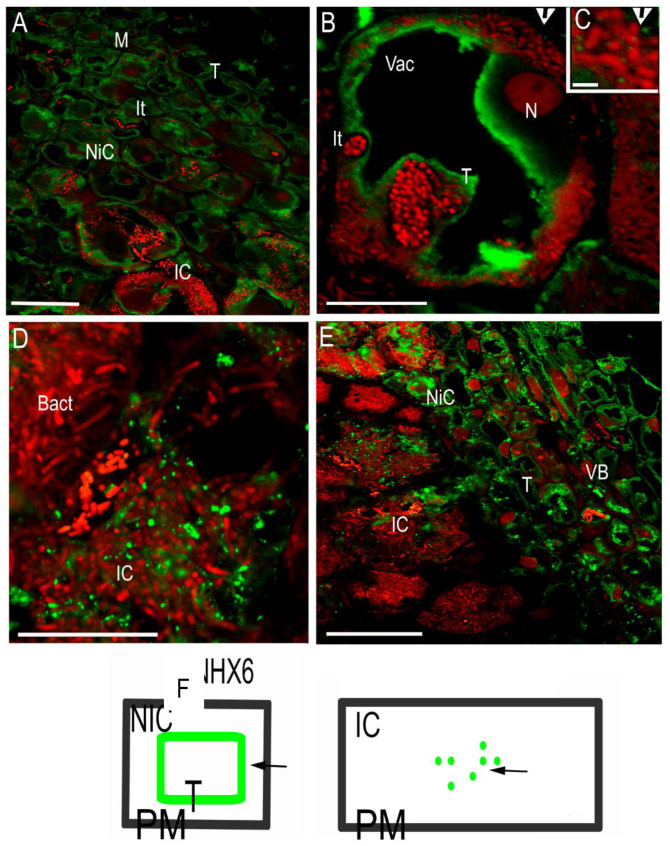
MtNHX6 (green fluorescence) was visualized by using a specific custom-made antibody in wt nodules. Color code: red fluorescence—propidium iodide counterstained bacteria and nuclei, green fluorescence—protein, labeling. For the immunolocalization of MtNHX6, an antibody developed in rabbit and a secondary antibody developed in goat were used. (**A**) In the meristem and distal infection zone, the signal is present over the cytoplasm and tonoplast of young vacuoles in non-infected and freshly infected cells. (**B**) Freshly infected young cell, note the strong signal on the tonoplast, and the small dots in the cytoplasm (arrowhead). (**C**) High magnification of a fraction of (**B**). The signal is in small dots, early endosomes. (D) (The mature infected cells are filled with symbiosomes with the signal represented as big dots. (**E**) The vascular bundle with the labeling over the tonoplast; note the infected cells with dot-like pattern of labeling. (**F**) The scheme of NHX6 protein distribution as per immunosignal (in green color) to explain the pattern and its changes in infected and non-infected cells. Bact: bacteroid, IC: infected cell, NiC: non-infected cell, M: meristem, It: infection thread, VB: vascular bundle, N: nucleus, T: tonoplast. Arrowhead: signal over the plasma membrane (PM). Bars: (**A**,**E**): 25 μm; (**B**): 12.5 μm; (**C**): 1.1 μm; (**D**): 10 μm.

**Figure 3 ijms-23-10618-f003:**
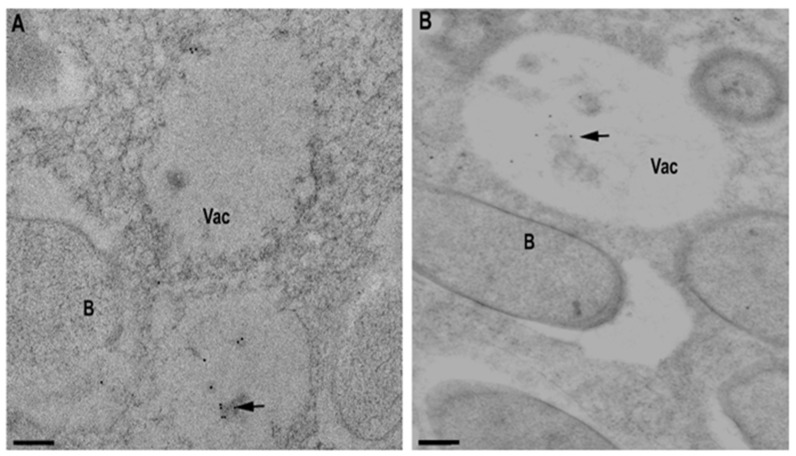
Electron microscopy immunolocalization of MtNHX7 (**A**) and MtNHX6 (**B**). Note the immunogold signal in the small vacuoles (~1000 nm). Bar: 200 nm. Vac: vacuole, B: bacteroid, arrow: immunogold signal.

**Figure 4 ijms-23-10618-f004:**
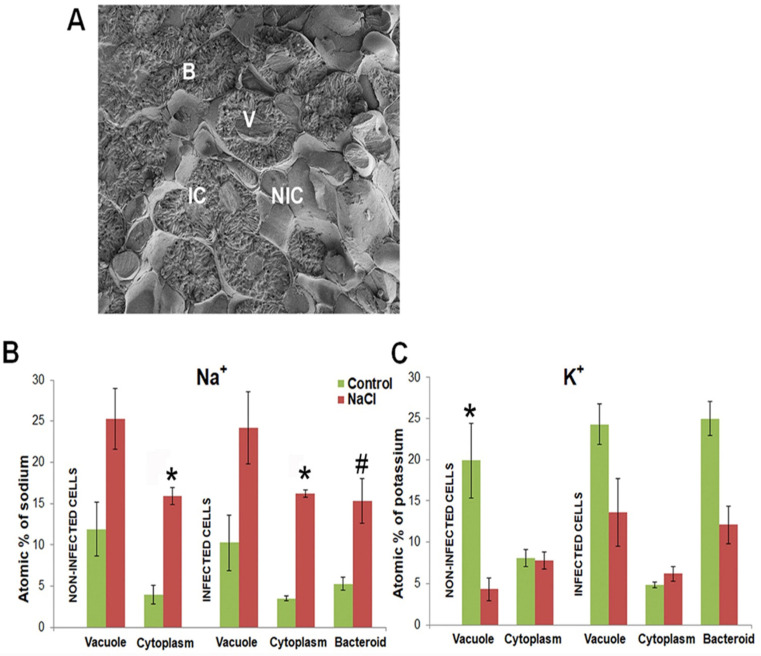
Scanning microscopy image of root nodule cells (**A**) and distribution of Na^+^ (**B**) and K^+^ (**C**) in cells of the nitrogen-fixing zone from control and salt-treated nodules of *M. truncatula*. B, bacteroid; IC, infected cell; NIC, non-infected cell; V, vacuole. (*) *p*-value ≤ 0.01, (#) *p*-value ≤ 0.05.

**Figure 5 ijms-23-10618-f005:**
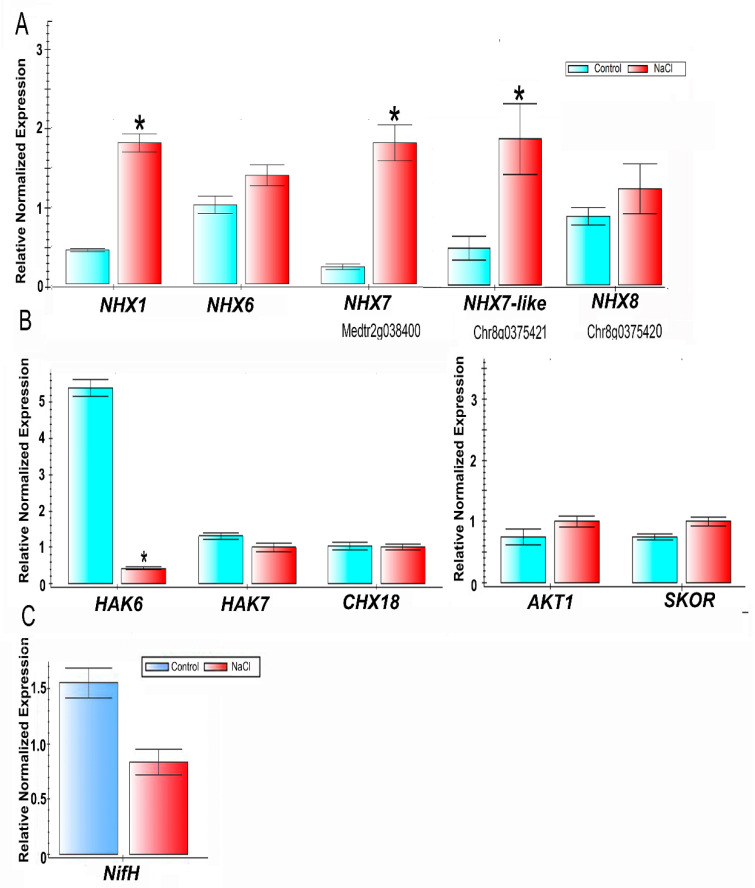
qPCR gene expression analysis of *M. truncatula* genes in nodules of control plants and nodules from the plants under salt stress. (**A**) the expression of *Na^+^*/*K^+^* (*NHX*) exchangers; (**B**) the expression of high-affinity K^+^ transporters 6 and 7, exchanger *CHX18* and K^+^ channels *AKT1* and *SKOR/GORK*; (**C**) expression of the *nifH* gene to estimate the nitrogen -fixing activity. Reference genes: *MtMtc27*, *MtGAPDH*, *SMc00128.* Asterisks (*) indicate significant differences between control and salt treatment.

**Figure 6 ijms-23-10618-f006:**
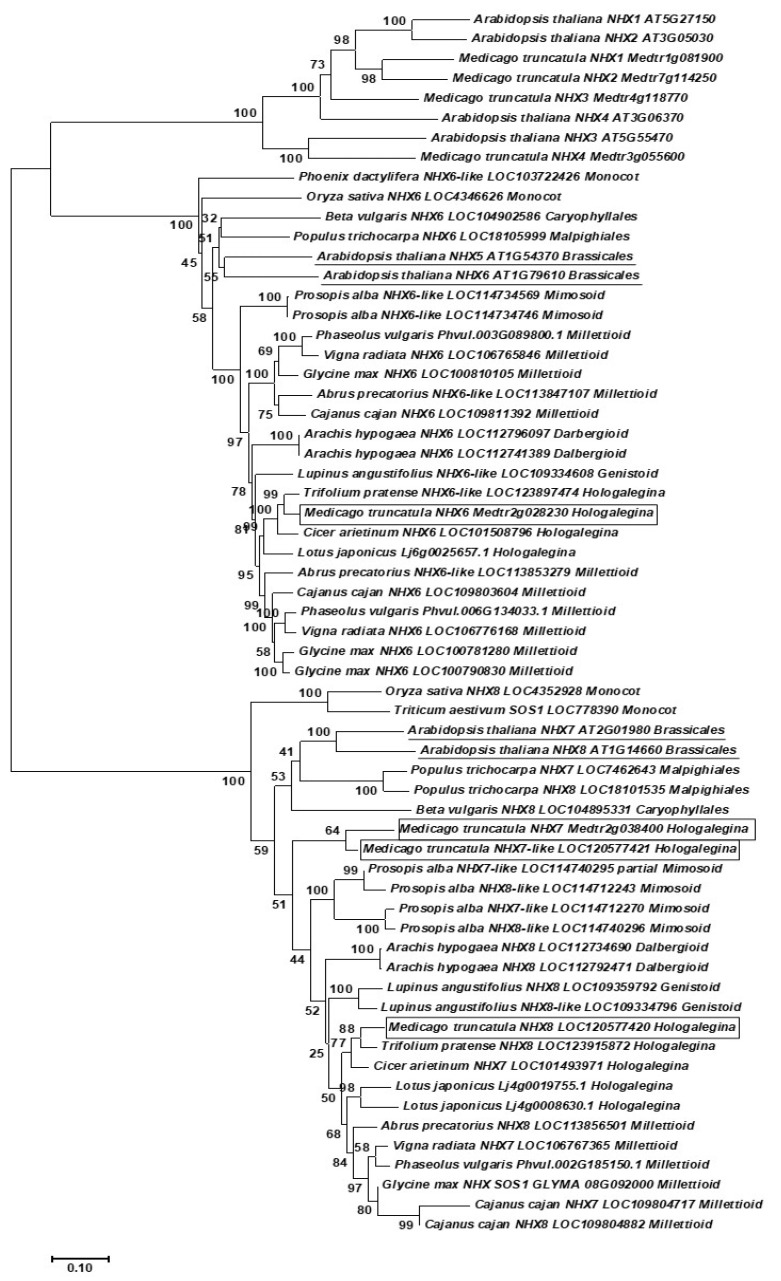
Evolutionary analysis of NHX genes in *M. truncatula*, other legume species, and *Arabidopsis thaliana*. Other plant species where NHX6-7-8 genes have been characterized also have been included. The *M. truncatula* NHX6-7-8 homologous genes are contained in boxes. The corresponding *A. thaliana* homologs are underlined. Bootstrap values are shown next to the branches (1000 pseudoreplicates). The tree is drawn to scale, with branch lengths measured in the number of substitutions per site. For each gene, the gene name is indicated. Plant orders (for non-legume species) and legume clades are also indicated.

**Table 1 ijms-23-10618-t001:** Expression of *M. truncatula* genes, encoding proteins of the inward transport of potassium and cation/sodium exchangers in the nodule developmental zones. Data (relative read distribution among zones (%)) were obtained from the Symbimics database. FI: Meristem; FIId: Distal infection zone; FIIp: Proximal infection zone; IZ: Interzone II–III; ZIII: nitrogen-fixing zone, mature zone.

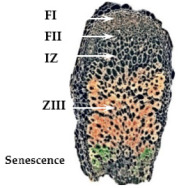						
	Nodule Zones	FI	FIId	FIIp	IZ	ZIII
Gene Name, ID or Locus Name						
** *Transporters* **						
***MtKT2***, *Medtr3g094090*		66.08 ± 2.6	17.27 ± 1.32	6.65 ± 1.96	6.37 ± 1.46	3.62 ± 0.9
***MtKT3***, *Medtr8g099090*		21.62 ± 1.22	19.49 ± 1.47	21.45 ± 0.94	19.81 ± 2.4	17.63 ± 1.11
***MtHAK5***, *Medtr4g099260*		49.65 ± 10.23	9.15 ± 2.14	28.75 ± 10.69	12.45 ± 5.25	0
***MtHAK6***, *Medtr5g034500*		1.97 ± 0.04	4.67 ± 0.37	9.82 ± 0.44	29.64 ± 2.57	53.9 ± 2.58
***MtHAK7***, *Medtr2g008820*		14.54 ± 1.22	16.21 ± 0.83	21.23 ± 1.51	19.61 ± 2.8	28.41 ± 3.83
***MtHAK8***, *Medtr6g007697*		60.39 ± 5.73	21.21 ± 3.33	14.44 ± 9.45	0.87 ± 0.44	3.09 ± 2.19
***MtHKT6***, *Medtr6g092840*		62.34 ± 5.45	34.62 ± 2.41	3.04 ± 3.04	0	0
** *Channels* **						
***MtAKT1***, *Medtr4g113530*		60.89 ± 8.22	5.16 ± 0.35	4.1 ± 3.19	16.73 ± 3.17	13.12 ± 1.99
***MtAKT2/3***, *Medtr2g006870*		38.76 ± 11.57	28.79 ± 18.34	19 ± 11.44	0	13.45 ± 13.45
***MtKAT1***, *Medtr8g446430*		39.29 ± 8.32	31.31 ± 7.11	17.04 ± 9.01	12.36 ± 6.65	0
***MtKAT3***, *Medtr3g108320*		47.12 ± 7.01	21.77 ± 7.94	1.14 ± 1.14	8.08 ± 4.04	21.89 ± 19.25
***MtSKOR/GORK***, *Medtr5g077770*		12.66 ± 1.86	9.25 ± 3.73	9.26 ± 6.19	44.47 ± 2.14	24.37 ± 5.38
***Exchangers* * **						
***MtCHX18***, *Medtr5g009770*		13.39 ± 0.97	12.69 ± 2.1	19.05 ± 3.61	29.47 ± 3.5	25.4 ± 2.55
***MtNHX1***, *Medtr1g081900*		14.34 ± 1.21	15.49 ± 1.64	21.25 ± 1.96	16.78 ± 0.31	32.14 ± 0.3
***MtNHX2***, *Medtr7g114250*		0	0	0	0	0
***MtNHX3***, *Medtr4g118770*		0	0	0	0	0
***MtNHX4***, *Medtr3g055600*		50 ± 13.89	43.05 ± 13.81	5.27 ± 5.27	0	1.67 ± 1.67
***MtNHX6***, *Medtr2g028230*		10.64 ± 0.85	21.25 ± 1.46	23.75 ± 1.93	20.94 ± 0.31	23.43 ± 2.23
***MtNHX7***, *Medtr2g038400*		40.08 ± 30.52	41.96 ± 29.97	17.96 ± 17.96	0	0
***MtNHX8***, *LOC120577420*		27.28 ± 2.61	21.11 ± 2.58	9.92 ± 4.22	17.41 ± 1.37	24.28 ± 2.37

*, Note that MtNH7-like is not available in the Symbimics database. Deseq-normalized RNA-seq reads are presented in Appendix A.

**Table 2 ijms-23-10618-t002:** Expression of potassium transporters/exchangers genes of *Sinorhizobium meliloti* in root nodule developmental zones. Data (relative read distribution among zones or organs (%)) were obtained from the Symbimics database. FI: Meristem; FIId: Distal infection zone; FIIp: Proximal infection zone; IZ: Interzone II–III; ZIII: nitrogen-fixing zone.

	Nodule Zones	FI	FIId	FIIp	IZ	ZIII
Gene Name, ID						
**Putative potassium transport system protein kup 1** (*SMc00873*)		10.3 ± 2.43	8.64 ± 0.8	11.21 ± 2.99	18.6 ± 1.63	51.24 ± 2.52
**Putative potassium transport system protein kup 2** (*SMa1798*)		4.84 ± 2.94	6.24 ± 0.93	2.74 ± 0.74	9.3 ± 1.68	76.88 ± 2.41
**Potassium-transporting ATPase KdpC subunit** (*RA1253, SMa2329*)		0	7.16 ± 3.6	7.56 ± 4.01	8.09 ± 1.27	77.19 ± 6.29
**Putative subunit A/B (pH adaptation potassium efflux system protein A/B, Pha system subunit A/B)** (*SMc03179*)		20.08 ± 4.89	24.19 ± 2.4	19.97 ± 4.57	19.88 ± 0.98	15.88 ± 1.01
**Na (+)/H (+) antiporter *NhaA*** **(Sodium/proton antiporter *NhaA*)** (SMa1913)		11.97 ± 0.98	19.36 ± 2.25	11.15 ± 2.27	16.15 ± 1.82	41.38 ± 1.88

Deseq-normalized RNA-seq reads are presented as Appendix A.

## Data Availability

Data are available from the corresponding author upon reasonable request.

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
