# Peer review of "Sodium Accumulation in Infected Cells and Ion Transporters Mistargeting in Nodules of Medicago truncatula: Two Ugly Items That Hinder Coping with Salt Stress Effects"

_ijms, 2022, doi:10.3390/ijms231810618_

Round 1
Reviewer 1 Report
In this paper authors propose a mechanism to explain why legume nodules are sensitive to salt stress based on the correlation between salt content in infected and non-infected cells and the expression and location of different Na+/K+ exchangers. Functional experiments such as the overexpression of NHXs or other K+ transporters in infected cells to make this organ more tolerant to salt would reinforce their hypothesis, although timewise these experiments are probably beyond the reach of this work.
There are several major comments for the authors:
Line 238-256: The phylogenetic analysis of a small subset of NHX proteins is not relevant enough to be a section. Even if the authors want to include this section in the main text, it should be presented at the beginning of the results before the transcriptomic analysis. The phylogenetic analysis should be repeated, including other functionally characterized members from different plant species. That would help to capture specific signatures in the transporters involved in salinity.
In silico data mining should include also the Gene Expression Atlas and other transcriptomes (especially relevant the comparison between infected and non-infected cells: Limpens et al., 2013. PLoS One). Also, authors should include in Tables 1 & 2 the actual number of reads to address the biological significance of each transporter.
On a related notice, why did the authors focused on NHX6 and NHX7? Are these the most abundant transporters? Are they induced by salinity in roots of +N grown plants (as in Arabidopsis)?
Line 185-190 (Fig 1; confocal): Do the authors have an overview (low magnification of the nodule (where the different zones appear and we can identify the temporal changes that are taking place. Figure leyends must be autoexplicative. In this case, authors should complete the figure legend including whether wt (as in the immunohistochemistry of MtNHX6) or transgenic plants were used (MtNHX7-GFP). Legend must include whether the colours represent. Model (figures J and K) should also be explained.
Authors should test whether salt stress changes the location of any of the transporters expressed in the M. truncatula nodule.
Other comments:
Lines 28-30: specify what is the correct location of these transporters.
Line 31: rephrase.
Line 117,123, 156, 187, 188, 240, 251 : Species names should be in italics
Line 136: Correct comma typo in MtNHX1, MtNHX6,),
Lines 156-160. Unclear, please rephrase because it is difficult to follow.
Line 163: remove "in contrast to young cells,".
Line 164-166: Please, indicate what those dot-like structures may represent. Ideally, I would like to see colocation with subcellular markers.
Line 192: Authors must declare what the material was in each case (whether immunohistochemistry vs Phusion reporter).
Line 212 (Figure 3): Homogenize format in asterisks. Complete figure leyend.
Lines 236-237 (Figure 4). Complete figure legend with reference genes used, time after the salt treatment...
Line 253: Correct typo "sup-port"
Lines 492-493: Please clarify the number of replicates. In case, represent data accordingly using individual points
Author Response
Comments and Suggestions for Authors
In this paper authors propose a mechanism to explain why legume nodules are sensitive to salt stress based on the correlation between salt content in infected and non-infected cells and the expression and location of different Na+/K+ exchangers. Functional experiments such as the overexpression of NHXs or other K+ transporters in infected cells to make this organ more tolerant to salt would reinforce their hypothesis, although timewise these experiments are probably beyond the reach of this work.
A: The authors are very grateful to the Reviewer1 for the help in improving the article and his(her) very useful recommendations. We have corrected the text of the article according to the recommendations of the Reviewer1
A:We agree that the overexpression can have positive results to remediate detrimental effects in root nodules. We are planning to use the overexpression of host and rhizobia potassium channels in the future experiments. But NHX genes, as we know now from our results, do not look like promising candidates for positive effect of overexpression, due to their rapid mistargeting in infected cells, that can be detected already in 2-3 cell layers of nitrogen fixation zone. We are thankful for Reviewer 1 for (her)his suggestions.
There are several major comments for the authors:
Line 238-256: The phylogenetic analysis of a small subset of NHX proteins is not relevant enough to be a section. Even if the authors want to include this section in the main text, it should be presented at the beginning of the results before the transcriptomic analysis. The phylogenetic analysis should be repeated, including other functionally characterized members from different plant species.
A:We have moved the phylogenetic analysis to the Material and methods section, as it was suggested by Reviewer 1. The phylogenetic analysis is expanded and now includes several homologs from other plant species (Oryza sativa, Phoenyx dactylifera, Triticum aestivum, Beta vulgaris, Populus trichocarpa. Lines :555-639
In silico data mining should include also the Gene Expression Atlas and other transcriptomes (especially relevant the comparison between infected and non-infected cells: Limpens et al., 2013. PLoS One).
The reference to Limpens et al,2013 is added to the List of References, the explication from the Supplementary of Limpens et al.,2013 is presented as Supplementary tableS4, here are listed genes of cation ion transporters differentially expressed in infected and non-infected cells in nitrogen-fixing zone. Data from Limpens et al.,2013 and Gene Atlas concerning the expression in zone of infection, and infected and non-infected cells of zone of nitrogen fixation are included to the text: Lines 141-143, 354-364
Also, authors should include in Tables 1 & 2 the actual number of reads to address the biological significance of each transporter.
The table with the numbers of reads added to SupplementaryTable5 and 6
On a related notice, why did the authors focused on NHX6 and NHX7? Are these the most abundant transporters? Are they induced by salinity in roots of +N grown plants (as in Arabidopsis)?
Two M.truncalula NHX exchangers MtNHX7 and MtNHX6, were used for immunolocalization analysis.
MtNHX7 is a homolog of AtNHX7(SOS1)gene, as well as it is homologous to NHX7 from other plants ( Oryza sativa, Phoenyx dactylifera, Triticum aestivum, Beta vulgaris, Populus trichocarpa) where it is induced by salt treatment and associated with salt tolerance (69,70,71,72). AtSOS1 is localized on plasma membrane and involved in Na+ exclusion from the plant cell via the plasma membrane to apoplast. Upregulation of NHX7/SOS1 has been associated with salt tolerance in M. truncatula and M. falcata [28,29], and other fabaceans [30].
The exchanger MtNHX6 is a homolog of AtNHX6 [20] and other plants used in Phylogenetic analysis. AtNHX6 a Golgi/endosome/PVC membrane resident, participates in K+or Na+sequestration into endosomal lumen[17,25]. MtNHX6, endosomal membrane exchanger, has been selected for the immunolocalization study due to known changes of symbiosome membrane identity and the appearance of some PVC endosome markers on the membrane during the symbiosis development [35]. But in presented study we haven’t found the relocation of this exchanger to symbiosome membrane. The expression of both genes in root nodules is induced by salt (Fig.5).
The explanation is added to the text , lines 164-175
Line 185-190 (Fig 1; confocal): Do the authors have an overview (low magnification of the nodule (where the different zones appear and we can identify the temporal changes that are taking place.
The image of root nodule, indicating nodule zonation is added to Table 1.
Figure leyends must be autoexplicative. In this case, authors should complete the figure legend including whether wt (as in the immunohistochemistry of MtNHX6) or transgenic plants were used (MtNHX7-GFP).
The explanations are added to Figure legend. Lines 173-174,185, 201
Legend must include whether the colours represent. Model (figures J and K) should also be explained.
The color code additional explanations are added to Figure legend (lines 187, 201-202.)
Authors should test whether salt stress changes the location of any of the transporters expressed in the M. truncatula nodule.
We did it for NHX6. The salt does not change the location of the transporter, does not show any specific effect on protein localization. Stress accelerates nodule senescence, that’s all. The images of immunolabelling after the salt treatment give the pattern similar to untreated nodules advanced to senescence stage, for instance the cells, situated to the basal-senescent part of the nodule. The protein signal was visible as big dots in non-infected cells, infected ones were near devoid of it. In salt stressed nodules some cells shown the signs of started plasmolysis, but that has been described years ago.
Lines 28-30: specify what is the correct location of these transporters.
Corrected, lines 29-30A
Line 31: rephrase.
The sentence was rewritten, lines 31-32
Line 117,123, 156, 187, 188, 240, 251 : Species names should be in italics
Corrected
Line 136: Correct comma typo in MtNHX1, MtNHX6,),corrected
Lines 156-160. Unclear, please rephrase because it is difficult to follow.
The paragraph has been rewritten : lines 164,165,166
Line 163: remove "in contrast to young cells,".
Corrected
Line 164-166: Please, indicate what those dot-like structures may represent. Ideally, I would like to see colocation with subcellular markers.
To specify the organelles we have performed the immunogold localization ( Fig.3), where it is shown that the protein signal is situated in the lumen of young vacuoles
Line 192: Authors must declare what the material was in each case (whether immunohistochemistry vs Phusion reporter).
Corrected, lines 185-186, 201
Line 212 (Figure 3): Homogenize format in asterisks. Complete figure leyend.
Corrected, figure legend added
Lines 236-237 (Figure 4). Complete figure legend with reference genes used, time after the salt treatment.
A To Figure legend added the names of reference genes line 304
:Constitutively expressed genes are named in Methods, line 487.
Line 253: Correct typo "sup-port"
Corrected
Lines 492-493: Please clarify the number of replicates. In case, represent data accordingly using individual points
Explained, line 645
Reviewer 2 Report
Dear Authors,
recently I had a privilege to review your manuscript "Sodium accumulation and ion transporters mistargeting in nodules of Medicago truncatula: two ugly items that hinder coping with salt stress effects" for "International Journal of Molecular Sciences".
The manuscript is interesting, provides novel results and it should be published after the necessary corrections. All my comments and requirements are indicated in the annotated manuscript that I attach to this review, and which is an integral part of the review.
Any environmental stress has many physiological effects, early and late, direct and indirect (excuse me for this truism), so studies of the effects of such stresses should take into account as broad a physiological background as possible in stressed plants. In the case of salinity, the water management of the plant should be taken into account, and in the case of studies on the fabacean root nodules, photosynthesis and the level of N2 fixation should be necessarily taken into account, as the N2 fixation efficiency depends directly on photosynthesis, and any drop in N2 fixation level would induce the plant to decrease the cost of nodule maintenance, which at the histological level is visible as degradative changes in the mature bacteroid-containing tissue. Such background was not "created" in the reviewed study, which makes it difficult to understand the biological significance of the observed phenomena. This does not mean that the presented analyzes are worthless, but it reduces their importance to the level of contributory studies.
It should also be noted that the salinity stress was maintained for 2 weeks, which means that the tested plants had already entered the stage of adaptation to stress. It would be appropriate to discuss this fact.
I am wondering about the sense of the subchapters order in the manuscript - it seems more natural to first prove that the plants were stressed indeed using some physiological indicators, next perform a broad analysis of gene expression (using a phylogenetic analysis of homologs to facilitate selection of the genes for expression analysis), and then to investigate in situ the expression of products of those genes whose expression pattern looks the most relevant/ interesting. However, in the Results, the reporting sequence is apparently, random ;-) and the criteria for protein selection for immunolocalization are unknown. The subchapter on phylogenetic analysis results seems a bit anachronistic as of today - such an analysis fits M&M rather than Results of the presented study.
The chapter on M&M is written, especially in some parts, in a very laconic way, thus failing to meet the postulate of the scientific method about repeatability of research. This chapter must be supplemented with all the missing details, as indicated in the manuscript file attached to this review.
The results of the microscopic analysis in the part concerning protein localization using a confocal microscope are poorly documented - in the sense that the presented microscopic documentation does not show the described features, because the images' magnifications and resolution are too low in the manuscript, and thus the details are not discernible. I suggest adding high magnification images to the relevant figure, which will make it possible to actually distinguish the described parts of the cell, especially the cell membrane or cytoplasm (=cytosol). The set of the immuno-tag and fluorochrome (PI) is insufficient. Some fluorochrome for the cell wall would make the immuno-images much more convincing - some of the images "suffer" from the impossibility to recognize cell boundaries. Repeating this analysis would be impractical or even impossible as it is time-consuming and costly, but, perhaps/ hopefully ;-) DIC or BF twin-images could be added to the figure to make the histological context better discernible.
Some of the Figure or Table legends are to brief - generally, it is expected that a legend is sufficient to understand the content of a figure or a table.
In the elemental analysis, the cell walls were omitted - why? The apoplast is a compartment, where the elements excluded from the cell or prevented to be imported could be accumulated. If you were able to discern the bacteroids and cell cytosol, the resolution of the measurements was absolutely sufficient to collect data from cell wall also.
Last but not least, the language and edition must be corrected (minor problems in the text and in the References).
With best regards,
Sincerely yours,
Reviewer

Author Response
Dear Authors,
recently I had a privilege to review your manuscript "Sodium accumulation and ion transporters mistargeting in nodules of Medicago truncatula: two ugly items that hinder coping with salt stress effects" for "International Journal of Molecular Sciences".
The manuscript is interesting, provides novel results and it should be published after the necessary corrections. All my comments and requirements are indicated in the annotated manuscript that I attach to this review, and which is an integral part of the review.
A: we are extremely thankful for Reviewer 2 for his comprehensive review of our paper and his suggestions to improve it. We accepted all correction that were indicated in PDF file, attached to ms (see attached text with corrections).
Any environmental stress has many physiological effects, early and late, direct and indirect (excuse me for this truism), so studies of the effects of such stresses should take into account as broad a physiological background as possible in stressed plants. In the case of salinity, the water management of the plant should be taken into account, and in the case of studies on the fabacean root nodules, photosynthesis and the level of N2 fixation should be necessarily taken into account, as the N2 fixation efficiency depends directly on photosynthesis, and any drop in N2 fixation level would induce the plant to decrease the cost of nodule maintenance, which at the histological level is visible as degradative changes in the mature bacteroid-containing tissue. Such background was not "created" in the reviewed study, which makes it difficult to understand the biological significance of the observed phenomena. This does not mean that the presented analyzes are worthless, but it reduces their importance to the level of contributory studies.
It should also be noted that the salinity stress was maintained for 2 weeks, which means that the tested plants had already entered the stage of adaptation to stress. It would be appropriate to discuss this fact.
Authors appreciate the suggested by Reviewer 2
Authors have added the paragraphs lines 306-310 describing the cost of nodule maintenance in terms of photosynthesis and the effect of salt. Two references related to the photosynthesis have been added :
- Hanin M., Ebel C.,Ngom M.,Laplaze L.,Masmoudi K.2016. New Insights on Plant Salt Tolerance Mechanisms and Their Po-tential Use for Breeding.Front Plant Sci., 29;7:1787. doi: 10.3389/fpls.2016.01787. eCollection 2016
- Booth, N.J.; Smith, P.M.C.; Ramesh, S.A.; Day, D.A. Malate Transport and Metabolism in Nitrogen-Fixing Legume Nodules. Molecules 2021, 26, 6876. https:// doi.org/10.3390/molecules26226876
The experiment design with prolonged salt application was aimed to emulate the nature conditions and the effect of salt for the nodules. Obviously in the soil the salt cannot be removed within 1 or 3 days, as it often presented in the literature.
The nodules from the second generation we have used for the analyses were pink and do not have visible green senescent zone
The photo of nodules is in attached file
I am wondering about the sense of the subchapters order in the manuscript - it seems more natural to first prove that the plants were stressed indeed using some physiological indicators, next perform a broad analysis of gene expression (using a phylogenetic analysis of homologs to facilitate selection of the genes for expression analysis), and then to investigate in situ the expression of products of those genes whose expression pattern looks the most relevant/ interesting.
A: Many works have been performed to estimate the effect of salt for the nodule, it was described the cytology showing the lysis and rapid nodule senescence, drop in nitrogenase activity, and some trascriptome changes. So the nodules are stressed. Salt causes rapid termination of symbiosis, the lysis of infected cells and it actually killing the nodule, by the same time cells of the adjacent root are suffering a bit, but not dying out. Why? We try to find the causes of the nodule fragility using the defined genes as instruments to find some differences between infected and non-infected cells in terms of ability of infected cells to defend itself in case of stress.
However, in the Results, the reporting sequence is apparently, random ;-) and the criteria for protein selection for immunolocalization are unknown.
A:We have added the paragraph (lines 164-175) describing the criteria for proteins selection
The subchapter on phylogenetic analysis results seems a bit anachronistic as of today - such an analysis fits M&M rather than Results of the presented study.
According to the suggestions of Reviewers Phylogenetic analysis is shifted to the Materials and methods. The phylogenetic analysis is expanded and now includes several homologs from other plant species (Oryza sativa, Phoenyx dactylifera, Triticum aestivum, Beta vulgaris, Populus trichocarpa). Lines :555-639
.
The chapter on M&M is written, especially in some parts, in a very laconic way, thus failing to meet the postulate of the scientific method about repeatability of research. This chapter must be supplemented with all the missing details, as indicated in the manuscript file attached to this review.
According to the suggestions of Reviewer 2 the details of methods were added to Material and methods lines 477, 482, 496-503, 510,
The results of the microscopic analysis in the part concerning protein localization using a confocal microscope are poorly documented - in the sense that the presented microscopic documentation does not show the described features, because the images' magnifications and resolution are too low in the manuscript, and thus the details are not discernible.
A: To improve the microscopic documentation, the figure 1 has been split in two, now images are presented as Fig.1 and Fig2., that helps to increase the size of the photos and resolution. However, the submission rules have some limitation for size of the files. Resolution is also reduced when PDF is created from tiff files.
I suggest adding high magnification images to the relevant figure, which will make it possible to actually distinguish the described parts of the cell, especially the cell membrane or cytoplasm (=cytosol).
The set of the immuno-tag and fluorochrome (PI) is insufficient. Some fluorochrome for the cell wall would make the immuno-images much more convincing - some of the images "suffer" from the impossibility to recognize cell boundaries.
A: According to the suggestions of Reviewer 2, we have added to Fig 1 additional image (B), high magnification of fractions of image(A). The high magnification permits to see on the cell boundaries two lines of fluorescent signal, labelling the plasma membranes of neighboring cells, and the cell wall region between them ( small arrows). The additional abbreviation (cell wall, cytosol) are added. ”
Repeating this analysis would be impractical or even impossible as it is time-consuming and costly, but, perhaps/ hopefully ;-) DIC or BF twin-images could be added to the figure to make the histological context better discernible.
To explain the cytological context to confocal figures the immunolabelling and the plasma membrane were marked, additional image with high magnification ( Fig.1B, has been created on panel Fig1
Some of the Figure or Table legends are to brief - generally, it is expected that a legend is sufficient to understand the content of a figure or a table.
In the elemental analysis, the cell walls were omitted - why? The apoplast is a compartment, where the elements excluded from the cell or prevented to be imported could be accumulated. If you were able to discern the bacteroids and cell cytosol, the resolution of the measurements was absolutely sufficient to collect data from cell wall also.
The analysis of cell wall and apoplast was not the scope of this paper. This paper was centered on the comparison of intracellular ion content in infected and non-infected cells and in bacteroids. Apoplast is a compartment, that is encompassing and connecting all cells in root nodule. It would be difficult to “cut out” apoplast of infected cell from the non-infected one. The ion transport in apoplast is quite interesting task for future research, especially under the salt stress. We are very thankful for Reviewer 2 for his suggestion to perform such analysis. However, it needs a very careful planning, and a high level of funding to pay for the analysis, that is not so easy now
Last but not least, the language and edition must be corrected (minor problems in the text and in the References).
The text has been edited by mother English speaker

Round 2
Reviewer 1 Report
Thank you very much for clarifying the points raised in my previous report.
I still have a minor comment:
Line 175-177: I propose the alternative redaction: Wild type and transgenic nodules carrying ProMtNHX7:MtMTNHX7:GFP construct were used for the localization study of MtNHX6 and MtNHX7, using anti-MtNHX6 and anti-GFP antibodies respectively. A secondary Alexa488-conjugated antibody was used in both cases.
Author Response
Authors are thankful for the comments of Reviewer1. We have corrected lines 175-177 as it was suggested by Reviewer1.
Reviewer 2 Report
The manuscript „Sodium accumulation and ion transporters mistargeting in nodules of Medicago truncatula: two ugly items that hinder coping with salt stress effects” by Trifonova et al. is acceptable in the version revised by the Authors. However, it contains some editorial errors (spaces missing, additional dots etc; line 46 Fabaceae should be corrected to Fabacean) that should be corrected before publications.
Author Response
Authors are thankful for the comments of Reviewer2. The ms was corrected according to the suggestions of Reviewer2.